# Immunoinformatics-Aided Analysis of RSV Fusion and Attachment Glycoproteins to Design a Potent Multi-Epitope Vaccine

**DOI:** 10.3390/vaccines10091381

**Published:** 2022-08-24

**Authors:** Hamza Arshad Dar, Fahad Nasser Almajhdi, Shahkaar Aziz, Yasir Waheed

**Affiliations:** 1Department of Bio and Brain Engineering, Korea Advanced Institute of Science and Technology (KAIST), Daejeon 34141, Korea; 2Department of Botany and Microbiology, College of Science, King Saud University, Riyadh 11451, Saudi Arabia; 3Institute of Biotechnology and Genetic Engineering, The University of Agriculture, Peshawar 25000, Pakistan; 4Office of Research, Innovation & Commercialization, Shaheed Zulfiqar Ali Bhutto Medical University (SZABMU), Islamabad 44000, Pakistan

**Keywords:** respiratory syncytial virus, multi-epitope vaccine, molecular modeling, molecular dynamics, vaccine candidate

## Abstract

Respiratory syncytial virus (RSV) usually causes respiratory tract infections of upper airways in infants and young children. Despite recent medical advances, no approved vaccine is available to control RSV infections. Therefore, we conducted an immunoinformatics study to design and evaluate a potential multi-epitope vaccine against RSV. Sequence-based analyses of the glycoproteins F and G revealed a total of eight CD8 T-cell and three CD4 T-cell epitopes after considering antigenicity, binding affinity and other parameters. Molecular docking analysis confirmed that these T-cell epitopes developed strong structural associations with HLA allele(s). By integrating these prioritized epitopes with linkers and a cholera toxin-derived adjuvant, a multi-epitope vaccine was designed. The developed vaccine was found to be stable, non-allergenic, flexible and antigenic. Molecular docking analysis revealed a striking mean HADDOCK score (−143.3) of top-ranked vaccine-TLR cluster and a Gibbs free energy change (ΔG) value of −11.3 kcal mol^−1^. As per computational immune simulation results, the vaccine generated a high titer of antibodies (especially IgM) and effector T-cells. Also, codon optimization and in silico cloning ensured the increased expression of vaccine in *Escherichia coli*. Altogether, we anticipate that the multi-epitope vaccine reported in this study will stimulate humoral and cellular responses against RSV infection, subject to follow-up experimental validation.

## 1. Introduction

Human respiratory syncytial virus (RSV) belongs to the Pneumoviridae virus family [1,2]. The genome size of this species is ~15.19 kb, and it contains a single-stranded RNA with a negative sense [3,4]. A total of two subtypes of RSV have been reportedly identified, i.e., RSV-A and RSV-B [5,6]. RSV exhibits unique characteristics that distinguish it from its family members. The RSV virion is formed when the viral nucleocapsid is enclosed by a lipid envelope, that in turn is generated from the host cell’s plasma membrane [7]. The RSV genome contains a total of ten genes, including those which encode fusion protein F and attachment glycoprotein G [7,8].

Using the F protein, the virus exhibits binding to the host cell surface [9]. After this successful interaction, RSV enters the host cells and forms “syncytia”. The virus surface-associated fusion and glycoproteins essentially drive the synthesis of neutralizing antibodies in hosts, thus confirming their involvement in pathogenesis [7].

The virion membrane contains a metastable, prefusion form of the functional F protein trimer. Each F1 subunit has a fusion peptide (FP) at the N-terminus, which is a segment of hydrophobic residues that inserts into the target membrane. Near the C-terminus of F1, the transmembrane (TM) domain mirrors the FP, and each is linked to a heptad repeat (HR) in the following order: FP-HRA-HRB-TM. The pre-HRA refolds and trimerizes into the lengthy HRA helix when triggered [10]. The G protein is the most variable RSV protein, giving it a valuable property for research on RSV evolution. It has two large mucin-like domains that border the core area and are highly variable in their sequences. These two regions may serve as substrates for O-linked glycan decorating rather than serving any particular sequence or purpose, since the total Ser and Thr content of these two regions is rather consistent [10].

RSV is known to be very contagious [11]. The virus is transmitted primarily through respiratory droplets, direct contact with an infected individual, or exposure to surfaces contaminated with the virus [12]. RSV infections can also reoccur, as the infections weaken the long-term immunologic memory [13]. Infants are the major victims of RSV infection. While in the majority of cases, respiratory tract infections of the upper airways are caused, infections in the lower airways, such as bronchiolitis, may also occur [14]. A few symptoms are commonly observed, such as fever, sore throat, mainly nasal purulent drainage, as well as blockage of the respiratory air canals [15]. In severe cases, complications may arise, resulting in respiratory disorders such as pneumonia, asthma and bronchiolitis. The cytopathy that is caused by RSV is rather limited. It is believed that damage to air canals is brought on by immune responses, and not caused due to virus-mediated cell lysis [2].

At the time of writing, only two FDA-approved drugs were currently available to treat RSV infection [16]. In order to prevent viral infections in vulnerable children, a monoclonal antibody palivizumab is used. The use of this licensed drug has reduced hospitalization cases by 50 percent in high-risk infants [17]. Meanwhile, ribavirin has been granted FDA approval for treatment purposes in severe cases, albeit with limited effectiveness [18,19]. Due to the absence of effective treatment or prevention options to cater to all RSV infections, the development of robust therapies such as vaccines is much desired all over the world.

Research developments in the field of immunoinformatics have put immense importance on the selection of conserved epitopes (antigen-associated factors of pathogens recognized by host immune machinery) on the pathogen’s antigen(s) for vaccine development [20,21]. The T-cell epitope after being presented by the relevant HLA on the target cell’s surface activates the receptor present on the T-cell surface, which proliferates and ensures the induction of specific immune responses [22]. Depending on the epitope presented on the MHC surface, either a CD4+ T cell or CD8+ T cell epitope vaccine could be developed. There are many advantages associated with the use of epitope-based peptide vaccines. As they do not contain the full pathogen, their safety concerns are limited [23]. Furthermore, they are economical to produce (cheap and involve less daunting procedures). It is possible to incorporate sugar analogs in vitro in epitope vaccines. The production process takes less time compared to conventional vaccines; moreover, it results in enhanced specificity and stability [24]. The underlying principle of epitopes-based vaccines helps focus protective immune responses on designated and specified antigenic determinants, limiting the induction of unwarranted or unwanted immunological responses [25]. In immunoinformatics research, it is desirable to identify epitopes that tend to associate with a variety of HLA alleles, and thus provide protective immunity in human populations globally [26]. In this study, we conducted comprehensive immunoinformatics investigations on the RSV F and G glycoproteins in order to obtain high-ranked epitopes. The prioritized T-cell epitopes were joined together to design a multi-epitope, and then evaluated using rigorous structural modeling, molecular dynamics and immune simulations.

When microorganisms break through the mucosal barrier, Toll-like receptors (TLRs) can recognize them and initiate adaptive immune responses [27]. TLR4 has been previously investigated to recognize the viral structural proteins and stimulate the production of inflammatory cytokines [28]. Besides, several studies have indicated the importance of TLR4 in forming a robust immune response against viral infection [29,30]. Therefore, we explored the interaction of epitope-based vaccine with TLR4 structure using molecular docking, molecular dynamics simulation and free energy-based analyses. We believe our study is a valuable addition to previous vaccine studies on RSV. Subject to experimental validation, this will be a significant step towards developing an anti-RSV vaccine.

## 2. Materials and Methods

### 2.1. Collection of the RSV Glycoproteins F and G Sequences

The reference sequence of the RSV genome (accession number NC_001803.1) was extracted from the GenBank database of NCBI. From its annotation, the amino acid sequences of both the F and G glycoproteins were collected in FASTA format.

### 2.2. Prioritization of T-Cell Epitopes in the RSV F and G Glycoproteins

The F and G glycoprotein sequences were fed into the NetMHC 4.0 server (http://www.cbs.dtu.dk/services/NetMHC/ accessed on 7 April 2021) in order to forecast strong CD8 T-cell binders using default settings [31,32]. HLA I alleles selected for prediction were HLA-A*1101, HLA-A*0101, HLA-A*0301, HLA-A*0201 and HLA-B*3501. Next, these epitopes were assessed for antigenicity properties as per VaxiJen 2.0 (http://www.ddg-pharmfac.net/vaxijen/ accessed on 8 April 2021) by specifying a 0.5 threshold [33]. Next, the immunogenicity scores of epitope peptides were determined using the IEDB resource [34]. Finally, the binding potentials of these epitopes with their respective allele(s) were analyzed by calculating their IC50 values, using the MHCPred 2.0 (http://www.ddg-pharmfac.net/mhcpred/ accessed on 12 April 2021) [35]. Only epitopes exhibiting strong binding potential (IC50 value < 100 nm) were prioritized. It was also confirmed that these 9-mer peptides do not show identity with any human genomic region using BLASTp against the human genome.

The F and G glycoprotein sequences were checked using the NetMHCIIPan 3.2 server (http://www.cbs.dtu.dk/services/NetMHCIIpan-3.2/ accessed on 16 April 2021) in order to identify strong CD4+ T-cell binders [36]. HLA-DRB alleles selected for this purpose were DRB1*0101, DRB1*0401 and DRB1*0101. Next, antigenicity evaluation was carried out on these epitopes by VaxiJen 2.0 by specifying a 0.5 threshold [33]. Finally, their core 9-mer peptides (identified by the NetMHCIIPan server) were analyzed for binding affinity with their putative HLA DRB allele(s) using MHCPred 2.0 (https://www.ddg-pharmfac.net/mhcpred/MHCPred/), developed by Guan, Pingping, et al., Edward Jenner Institute for Vaccine Research, Berkshire, UK (accessed on 25 April 2021) [35]. Only those epitopes showing an IC50 < 100 nm were prioritized. It was also confirmed that these 15-mer peptides do not show identity with any human genomic region using BLASTp against the human genome. Moreover, the solvent accessibility of shortlisted epitopes residues was predicted via the WESA meta-predictor (https://pipe.rcc.fsu.edu/wesa/ accessed on 6 August 2022) [37].

### 2.3. Molecular Docking of Epitopes with HLA Alleles

The structural modeling of finalized T-cell epitopes was performed by the PEP-FOLD 3.5 online tool, specifying default settings [38]. Only the structure showing the lowest sOPEP energy was selected. The obtained epitope structures were used to conduct molecular dockings with their putative interacting HLA allelic structures that were defined earlier. This was necessary in order to perform structural characterization of the epitope-HLA interaction at the molecular level. We used the HADDOCK 2.2 server (http://haddock.science.uu.nl/services/HADDOCK2.2/), developed by Van Zundert, et al., Utrecht University, the Netherlands (accessed on 23 May 2021). This server was used for molecular docking analysis, and in order to calculate the various statistical properties associated with the overall process [39,40]. Guru interface, an advanced interface, was utilized for this purpose with its default settings.

The structures of complete HLA with co-crystallized peptides (HLA A*0101, HLA A*0201, HLA A*0301, HLA A*1101, HLA DRA/B*0101 and HLA DRA/B*0401) were retrieved from the RCSB database. These structures were retrieved from PDB ID 4NQX, 1DUZ, 3RL1, 1Q94, 1AQD and 5JLZ, respectively [41,42,43,44,45,46]. Co-crystallized peptides containing the structure of HLA alleles were submitted to the PDBsum server to identify those residues of HLA that formed bonded and non-bonded interactions with the co-crystallized peptides [47]. These amino acid residues were fed into the HADDOCK 2.2 server as the active residues of HLA molecules to guide the process of molecular docking. In the case of HLA I 9-mer peptides, the epitope residues were fed as active residues. However, for 15-mer HLA II epitopes, their core 9-mer regions were defined as active residues, whereas their surrounding residues were automatically designated as passive. The top cluster having the minimum HADDOCK score was considered, and a representative protein-peptide complex from it was downloaded in PDB format and visualized in UCSF Chimera version 1.15, developed by Resource for Biocomputing, Visualization, and Informatics (RBVI), University of California, San Francisco, CA, USA. [48]. The active binding site residues of HLA were identified using the PDBsum analysis suite [47]. The protein-peptide complexes obtained were also visualized using the UCSF Chimera for better representation [48].

### 2.4. In Silico Design of RSV Multi-Epitope Vaccine

A multi-epitope vaccine was designed against RSV by incorporating all the prioritized T-cell epitopes. Using a special linker GPGPG, epitopes were joined together. Meanwhile, cholera toxin subunit B was placed towards the N-terminal end of epitopes as an adjuvant to enhance the overall immunogenicity of the vaccine. The adjuvant was connected to the first epitope by a single rigid linker EAAAK. Crucial chemical and physical properties of the designed multi-epitope were studied using the ProtParam server’s Expasy tool [49]. These predictions concerned the half-life, molecular weight pH, hydropathicity, instability and aliphaticity. Furthermore, the designed poly-epitope vaccine was subjected to antigenicity checks by VaxiJen and AntigenPro servers [33,50]. In order to confirm that the vaccine does not trigger unwanted allergenic responses, AllergenFP version 1.0 and AllerTOP version 2.0 (both tools developed by Dimitrov, Ivan, et al., from Medical University of Sofia, Sofia, Bulgaria) were also used [51,52].

### 2.5. Structural Modeling of the RSV Multi-Epitope Vaccine

Three-dimensional modeled structures of the multi-epitope vaccine were generated by multiple servers: Phyre2, RoseTTAFold, i-TASSER, RAPTORX and 3Dpro [53,54,55,56,57]. All the structures that were obtained were checked for stereochemical quality and structural correctness by PROCHECK, ERRAT and ProSa-Web [58,59,60,61]. The top three modeled structures were subjected to molecular refinements by the GalaxyRefine web server (http://galaxy.seoklab.org/ accessed on 27 May 2021) [62]. Careful comparison of structures obtained before and after molecular refinements helped to select one model. In particular, Ramachandran plot analyses and ERRAT scores played a crucial role in model selection.

### 2.6. Molecular Docking Analysis of Vaccine with Toll-Like Receptor 4

The full-chain sequence of human TLR4 (without any signal peptides) was obtained from the UniProt sequence data in FASTA format. NCBI PSI-BLAST search was performed against this protein sequence against the Protein DataBank database. TLR4 structure with PDB ID 3FXI was downloaded from the PDB database [63]. All molecules except the TLR chain were removed using the UCSF Chimera tool [48]. The UCSF Chimera built-in option of ‘Dockprep’ was used with its default settings to prepare the TLR4 and multi-epitope structures before docking. For performing molecular docking, the HADDOCK 2.2 server was used [39,40]. This web server requires active and passive residues of input protein structures; these residues were identified by the CPORT online tool with a very sensitive option, as it is recommended by the HADDOCK server [64]. The HADDOCK refinement interface was used to improve the overall positioning and docking pose of the vaccine-TLR4 docked structure [40]. Some notable interface residues within the vaccine-TLR4 complex and inter-chain bonding forces were determined by the PDBsum [47]. Thermodynamics-based calculation of the Gibb’s free energy change was calculated by the PRODIGY server (http://wenmr.science.uu.nl/prodigy/ accessed on 1 June 2021), using its default parameters [65].

### 2.7. Molecular Dynamics Analysis on Vaccine-TLR4 Complex

GROMACS, a popular command line-based Linux tool, was used for molecular simulations on the vaccine-TLR4 complex [66]. Molecular simulations are usually recommended to imitate the environment in which biological interactions occur. This procedure also evaluates the stability of individual protein chains. Optimized Potential for Liquid simulation force field was selected [67]. The protein structural files were converted to gro format, in order to prepare the topology that is fully compatible with the chosen force field. In order to accommodate the water molecules, the vaccine-TLR4 structure was placed within the center of a cubic box having a 1-nanometer distance from the edge so that the structures’ periodic images were 2 nm apart. Water molecules were position-restrained using a constant force of 1000 kJ mol^−1^ nm^−2^. In order to neutralize the overall charge distribution in the system, the genion tool was used to add counter ions into the system. For this purpose, a Verlet cut-off scheme as well as Ewald electrostatic forces were applied.

The process of energy minimization was completed at 2098 steps. Graphs obtained during various stages of simulation and/or equilibration steps, such as energy minimization, temperature, pressure and density, were analyzed using Xmgrace tool version 5.1.19 (developed by Paul J. Turner, Center for Coastal and Land-Margin Research Oregon Graduate Institute of Science and Technology, Beaverton, OR, USA.) [68]. MD simulation was performed on the equilibrated vaccine-TLR4 structure for 20 ns. RMSD analysis concerning the backbone of the stabilized structure was carried out. During an MD production run of 20 ns, the radius of gyration parameter was also calculated to understand the compactness of the vaccine-TLR4 structure. Finally, the RMSF of side chains was also computed to check the flexibility of these regions during simulation.

### 2.8. In Silico Immune Simulation Analysis

In order to conduct computer-aided investigations on the immune potential of the vaccine, the C-ImmSim (http://kraken.iac.rm.cnr.it/C-IMMSIM/ accessed on 12 August 2021) server was used [69]. By combining a position-specific scoring matrix with machine learning, this server assesses epitope prediction and dynamic immunological activities. The whole analysis was performed using the protocol previously reported [70,71]. Briefly, three in silico doses were administered four weeks apart, at time steps of 1, 84, and 168 (single time step equals 8 h of daily life), respectively, for a total of 1050 simulation steps (about 12 months). Nevertheless, the default values of remaining parameters were used.

### 2.9. Cloning of Designed Vaccine

Using the EMBOSS Backtranseq platform, a reverse translation of the vaccine sequence was conducted [72]. Next, in order to optimize the codon sequence, the Java Codon Adaptation Tool at http://www.jcat.de/ (developed by Grote, Andreas, et al., Technische Universität Braunschweig, Germany), was utilized (accessed on 14 August 2021) [73]. This server provides Codon Adaptation Index (CAI) and GC% content, which were checked to evaluate expression levels of the vaccine. In addition, *Xho*I and *Nde*I restriction sites were added at the N-terminal and C-terminal ends of the optimized codon sequence of the vaccine candidate, respectively. Finally, the SnapGene tool was used (accessed on 16 August 2021) to clone the optimized sequence of vaccine into the expression vector pET28a (+) between *Xho*I and *Nde*I loci.

## 3. Results

### 3.1. Prioritization of T-Cell Epitopes in the RSV F and G Glycoproteins

Initially, 42 epitopes were predicted in the F and G glycoproteins that showed a strong putative association with HLA I alleles. Out of these, it was found that a total of 25 epitopes showed good VaxiJen scores (>0.5), thus indicating their antigenic tendencies. Immunogenicity evaluation of these sequences shortlisted a total of 13 epitopes. Finally, out of these, it was revealed that a total of eight epitopes had IC50 values < 100 nm, which indicates their strong binding potential with the predicted HLA I alleles (Table 1). BLASTp analyses of these peptide sequences with the human genome revealed that they do not show identity with any human genomic region.

Initially, 49 epitopes were predicted in the F and G glycoproteins that showed a strong putative association with HLA II alleles. Out of these, a total of 28 epitopes showed good VaxiJen scores (>0.5), thus indicating their antigenic tendencies. After exploring their binding affinities in terms of IC50 values of their 9-mer core regions, as well as selecting the best epitopes from among the overlapping epitopes, a total of three CD4+ T-cell epitopes were prioritized accordingly (Table 2). We also included AIIFIASANHKVTLT epitope in the DRB1*0401 results obtained, although it had an IC50 value somewhat higher with that allele, i.e., 264.24, as it had a good antigenicity score and binding affinity with another frequent HLA allele, DRB1*0101. Upon BLASTp analyses of these peptide sequences with the human genome, it was found that none showed identity with any human genomic region.

The selected T-cell epitopes showed good structural interactions with their cognate HLA allele(s) as indicated by their low HADDOCK scores (Appendix A). Among the HLA I T-cell epitopes, the lowest HADDOCK score (−113.8) was shown by epitope KTKNTTTTK, followed by epitope LLHNVNAGK (−104.2). Figure 1 shows interactions of top-ranked HLA I epitopes with their cognate HLA allele(s). HLA II T-cell epitopes also showed good structural interactions with their cognate HLA II allele(s) (Appendix A). Notably, the epitope AIIFIASANHKVTLT showed striking HADDOCK scores of −117.4 and −117.1 with HLA DRA/B1*0101 and HLA DRA/B1*0401, respectively. Best protein-epitope docked conformations can be visualized in Figure 2. These figures also visualized strong peptide-protein interactions within the binding groove of HLA molecule that is helpful for stable binding.

### 3.2. In Silico Designing, Physiochemical and Immunological Properties Evaluation of Multi-Epitope Vaccine

The prioritized HLA II epitope LIAVGLLLYCKARSTPVTLS contained the smaller HLA I epitope LIAVGLLLY; hence, the larger HLA II epitope was retained out of these two epitopes. In addition, overlapping residues were observed between two epitopes (HLA II epitope LGFLLGVGSAIASGI and HLA I epitope IASGIAVSK), so they were merged together to form one contiguous epitope LGFLLGVGSAIASGIAVSK. By joining all the prioritized T-cell epitopes with adjuvant and linkers (GPGPG and EAAAK), a multi-epitope vaccine was designed. As per the physicochemical analysis of the vaccine, its molecular weight is 26.657 kDa. It is generally acknowledged that proteins with a lower molecular weight than 110 kDa are more likely to show antigenic capabilities [21,74]. It was predicted that the in vitro half-life of the vaccine will be 30 h in mammalian reticulocytes, the in vivo half-life will exceed 20 h in yeast, and the in vivo half-life will exceed 10 h in *Escherichia coli*. As per server calculations, the instability index of protein is 26.61 (<40), thus stability is anticipated. A high value of 93.81 for the aliphatic index was obtained, which indicates increased thermostability at varying temperatures. A GRAVY score of 0.003 was obtained; thus, it was concluded that the vaccine is slightly prone to being hydrophobic. Therefore, the use of a stable, lipid-based reconstitution system would be helpful to overcome the protein’s self-assembly [75,76].

In considering all these results, it is expected that the multi-epitope vaccine meets the physiochemical requirements for production. Antigenicity evaluation by AntigenPro as well as VaxiJen servers revealed that the designed vaccine is antigenic. Allergenicity check by both AllergenFP and AllerTOP suggested the non-allergic tendencies of the vaccine, thus it is unlikely that the vaccine will trigger allergic responses in humans. The proposed vaccine’s amino acid sequence is provided within Appendix A.

### 3.3. Structural Modeling of RSV Multi-Epitope Vaccine

Upon careful analysis of structural models initially obtained by all methods, it was found that the RoseTTAFold server yielded better results. RoseTTAFold generated a total of five models. Out of these, one model (MODEL 05) was selected that was considered to have the best stereochemistry quality by the Ramachandran plot and high ERRAT scores (Table 3). However, it was observed that molecular refinements by the Galaxy Refine server did not improve the overall stereochemistry of the vaccine structure. Thus, MODEL 05 was shortlisted for downstream structural analysis. Figure 3 shows the structural evaluation results of MODEL 05. These evaluations were useful to gain crucial insights into structural modeling output. The best structural model was needed for continued structural analysis, including molecular docking, molecular dynamics simulations and so on. The 3D structure of the multi-epitope vaccine can be visualized in Figure 4.

### 3.4. Interaction Analysis of Vaccine with Toll-Like Receptor 4 by Molecular Docking

By analyzing docking results of vaccine and TLR4 structures, it was found that the most reputable cluster showed a HADDOCK score of 21.7 ± 4.3. After applying molecular refinements on the representative structure of this complex and clustering, a rapid improvement in the HADDOCK score was noticed. The refined vaccine-TLR4 complex showed a strikingly high HADDOCK score of −143.3 ± 9.6. A negative HADDOCK score signifies a greater potential of favorable interaction between the proteins. The statistical properties of this cluster, along with the HADDOCK score, are provided in Appendix A. A representative structure of vaccine-TLR4 was visualized, and interacting residues were colored. According to PDBsum’s detailed structural characterization, a total of 21 residues of vaccine formed associations with 25 residues of TLR4, with a total of 135 non-bonded contacts, 13 hydrogen bonds and four salt bridges. These interacting chains and interesting molecular interactions can be visualized clearly in Figure 5.

A computational assessment based on Gibbs free energy was conducted on the vaccine-TLR4 complex obtained after docking refinement. This provided crucial information about the binding affinity between the two protein chains. Through this analysis, it was revealed that the docked complex had a −11.3 kcal mol^−1^ ΔG value. The negative sign in Gibb’s free energy results strongly indicates that the association between multi-epitope and TLR4 is biologically feasible in terms of thermodynamics. Moreover, the dissociation constant Kd (M) value at 25.0 °C was calculated and found to be 5.6 × 10^−9^.

### 3.5. Molecular Dynamics Analysis on Vaccine-TLR4 Complex

The GROMACS tool analyzed the stability of TLR4 and the multi-epitope ligand complex. A total of 50,000 energy minimization steps were conducted till the system reached <1000 kJ/mol maximum force. Potential energy (E_pot_) was calculated. Ideally, E_pot_ value should be negative and in the range 10^5^–10^6^ for protein simulation in water. It was found that the E_pot_ value is −4.42718 ×10^6^ kJ mol^−1^ nm^−1^, whereas the 4370 atom faced the maximum force of 9.7368 × 10^2^ kJ mol^−1^ nm^−1^ (Figure 6A). Upon heating, the system’s temperature rapidly approached 300 K and showed minute fluctuations overall during equilibration (Appendix A). NPT equilibration was performed, and it was found that the average value of pressure was 1.6857 (Appendix A). The system’s average density was calculated to be 1014.57 kg/m^3^ as per NPT analysis (Appendix A). Mild fluctuations were observed in the protein backbones-based RMSD plot, indicating stability interactions between the two chains (Figure 6B). At 20 ns, the RMSD value approached ~0.4 nm. The RMSF plot (Figure 6C) associated with the protein side-chains showed various peaks, which indicated that many residues within the vaccine-TLR4 complex were highly flexible. The radius of gyration plot (Figure 6D), meanwhile, showed some fluctuations but mainly remained between 3.7 and 4.0 nm throughout the 20-nanosecond simulation. Altogether, these investigations supported a high level of structural stability within the vaccine-TLR complex.

### 3.6. Immune Simulation Analysis

After three in silico doses of the vaccine, various graphs were obtained (Figure 7). Simulated results showed that a high concentration of IgM was noted as a primary response. In addition, the significantly higher level of antibodies activities, including IgM, IgG1 + IgG2, as well as IgM + IgG antibodies in secondary and tertiary responses, were observed. Consequently, rapid clearance of antigen (lower concentration of antigen) was observed (Figure 7A). The vaccine structure induced a robust, long-established immunological response, as seen by the increased quantities of simulated B cells and memory B cell development (Appendix A). T-cell numbers, on the other hand, progressively declined following their initial rise with relative immunity development, which is required to elicit immunological responses (Figure 7B,C and Appendix A). Following the vaccine injections, macrophage activity improved, whereas dendritic cell activity remained stable (Appendix A). Similarly, higher levels of cytokine IFN-ϒ (>400,000 ng/mL) were noted (Figure 7D). The elicited immune features suggest that the vaccine could induce highly specific immune responses needed to fight the viral infection.

### 3.7. Optimized Cloning of the RSV Vaccine Candidate

Upon optimizing the codon sequence of the RSV vaccine, the length of the cDNA sequence was found to be 771 nucleotides (Appendix A). The obtained CAI (0.97) revealed that the adapted sequence is composed of codons suitable for using the target organism’s cellular machinery. Besides, the GC content (52.39%) after codon optimization was found to be acceptable. Therefore, ease of expression is guaranteed inside the *E. coli* K12 strain. Towards the N and C termini of codon-optimized nucleotides, the restriction sites of *Xho*I and *Nde*I enzymes were conjugated, respectively (Figure 8). In order to clone the vaccine, the final optimized sequence was placed within the pET28a (+) vector between the *Xho*I (158) and *Nde*I (936) positions. The cloned plasmid had a final length of 6607 bp.

## 4. Discussion

Vaccination is the perfect health strategy to resolve the menace of infectious diseases, as evidenced by the eradication of the smallpox disease. In this study, our research group explored the concept of an epitope-based anti-RSV vaccine, as opposed to the use of the whole pathogen. Subunit-based vaccines only comprise antigenic regions of the virulent pathogen; thus, they are considered safe for eliciting the appropriate immune responses [77]. Recent progress made in immunoinformatics has enabled the development of vaccines with limited time and resources, which is an encouraging development. However, the success of these computer-guided analyses is dependent on the accurate identification of good antigens and their molecular determinants (epitopes), as well as the efficiency of the delivery system. Altogether, the identification of good vaccine targets is essential in vaccine studies.

Vaccines that are based on epitopes are known to show good responses. The approach has been followed by other researchers to target various diseases, ranging from cancer to viral and other pathogen-related diseases [21,78,79,80]. At the forefront of these technologies are numerous immunoinformatics methods and tools that have significantly aided the proper identification of T-cell interacting epitopes. Vaccines based on T-cell epitopes are known to evoke durable immune responses and overcome the antigenic drift that is typically associated with evasion of antibody memory responses [26]. The T-cell subtypes (CD8+ and CD4+) play an important role in triggering antiviral responses and B-cell clonal expansion. Thus, we focused exclusively on the identification of reliable T-cell epitopes, using both sequence-based as well as structural bioinformatics analysis. We also designed a structurally stable multi-epitope vaccine using top-ranked T-cell epitopes, and elaborated on the vaccine’s effectiveness using molecular docking and molecular dynamics simulation with TLR4.

Epitope-based vaccines have drawn significant attention over classical immunization because of their several advantages [81]. First, the distinctive design strategy differentiates the epitope-based vaccines from single-epitope or conventional vaccines: they include multiple MHC-restricted epitopes that T-cell receptors can recognize from various T-cell subsets. They can stimulate simultaneous cellular and humoral immune responses due to the inclusion of B-cell and T-cell (CTL and HTL) epitopes. Some special substances, i.e., adjuvants, are added to increase immunogenicity and produce a long-lasting immunological response. Finally, undesirable substances are excluded, thus decreasing the chance of evoking adverse or harmful immune reactions [82]. Therefore, with such benefits, elegant epitope-based vaccines should progress to promising therapeutic options for viral infection. Using the immunoinformatics approach, the multi-epitope vaccine has been designed in recent years for several viruses, including chikungunya, dengue, hepatitis C, MERS-CoV, SARS-CoV-2 and Zika [81,83,84]. Also, several multi-epitope vaccines have already progressed to clinical trials [84]. Molecular dynamics analysis also enabled us to understand the conformational dynamics of the vaccine. It has been reported that regions of increased conformational order are thought to be more antigenic [85]. Considering the significant health burden of RSV infection, a prophylactic vaccine against the virus is desirable.

In this study, we aimed to assess the antigenic potential of two very important glycoproteins of RSV: fusion F and attachment G. Due to their role in viral attachment and entry, they are deemed to be crucial for RSV pathogenesis [10]. On top of that, they are the only RSV proteins known to target the human cell membrane and contain conserved regions. Therefore, we considered it prudent to choose them as targets for developing a potent anti-RSV vaccine.

Adverse effects may occur after administering vaccines, and could be due to any vaccine constituents [86,87]. The probability of generating harmful autoimmune responses cannot be ruled out [88]. For example, administering of hepatitis B vaccine may lead to thrombocytopenia and/or polyarthritis [89]. Similarly, vaccines against tetanus and rabies diseases could potentially cause neural complications [90]. In this study, the prioritized epitopes were checked through BLASTp analysis against the human genome, in order to determine whether they have autoimmune potential or not. Our analyses suggested that these epitopes are not anticipated to trigger autoimmunity, thus rendering them safe for use.

While bioinformatics tools are known to yield good results, one major issue faced by researchers is that the number of epitopes obtained after in silico filters is often quite high [91]. It is advisable to prioritize only the best and most promising epitopes out of the pool of epitopes obtained so that other scientists can validate these results by wet laboratory approaches. In order to reduce the epitope list to a manageable level, we applied filters such as antigenicity, non-homology to humans, immunogenicity and good HLA binding affinity. Moreover, we explored the structural associations of these finalized T-cell epitopes with their cognate HLA allele(s) using molecular docking. For this purpose, we used the HADDOCK 2.2 server, a highly robust and effective tool to conduct protein-peptide docking. Docking results revealed a strong association between the epitopes and their corresponding HLA allele(s), which further validated our approach.

Prioritization of epitopes by adopting an in silico sequence-based characterization and molecular docking analysis is a significant step towards the development of peptide-based vaccine candidates. However, more follow-up confirmatory experiments are crucial to validate the effectiveness of the epitopes as well as the multi-epitope vaccine against RSV. Nevertheless, the immunological data obtained in this study can be integrated with data from other databases related to genomics, proteomics or pharmacogenomics. By doing this, the scope and breadth of immunoinformatics analysis presented in this study can be increased.

## 5. Conclusions

In this study, our research group adopted a specialized immunoinformatics workflow to develop a multi-epitope subunit vaccine containing highly antigenic T-cell epitopes of the RSV glycoproteins F and G. Elaborate investigations using molecular docking, thermodynamic investigations, molecular dynamics and in silico simulation analyses strongly support the ability of our vaccine to not only associate strongly with Toll-like receptor 4, but also to trigger highly specific cellular and humoral immune responses against RSV pathogens. Altogether, the present study shows that bioinformatics approaches can help develop effective treatments for emerging viral infections, despite limited resources and time. However, these results are purely computational and need to be corroborated by wet laboratory methods.

## Figures and Tables

**Figure 1 vaccines-10-01381-f001:**
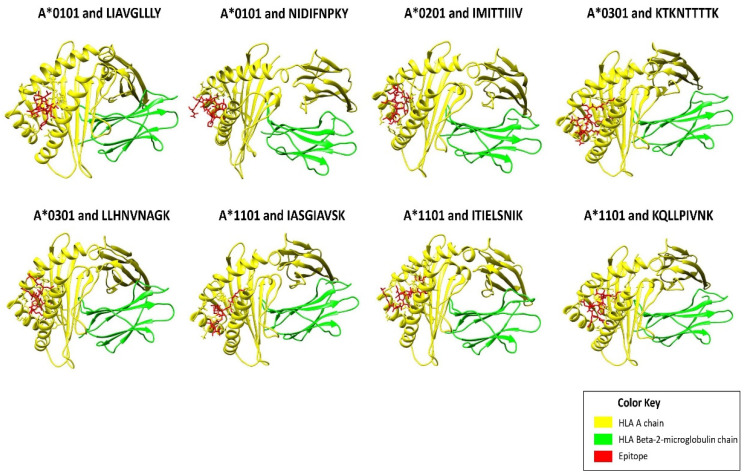
Figure showing structural interactions between the prioritized HLA I epitopes (colored in red) and cognate HLA I chains (colored in yellow and green). Out of all the HLA I epitopes, KTKNTTTTK (from the G glycoprotein) showed the best docking score (−113.8) with cognate HLA I molecule HLA-A*0301, followed by epitope LLHNVNAGK (−104.2) of the F glycoprotein with HLA-A*0301, and so on. It can be observed that all the epitopes fit comfortably inside the binding groove of the HLA molecule and thus are projected to interact strongly with their cognate HLA I allele.

**Figure 2 vaccines-10-01381-f002:**
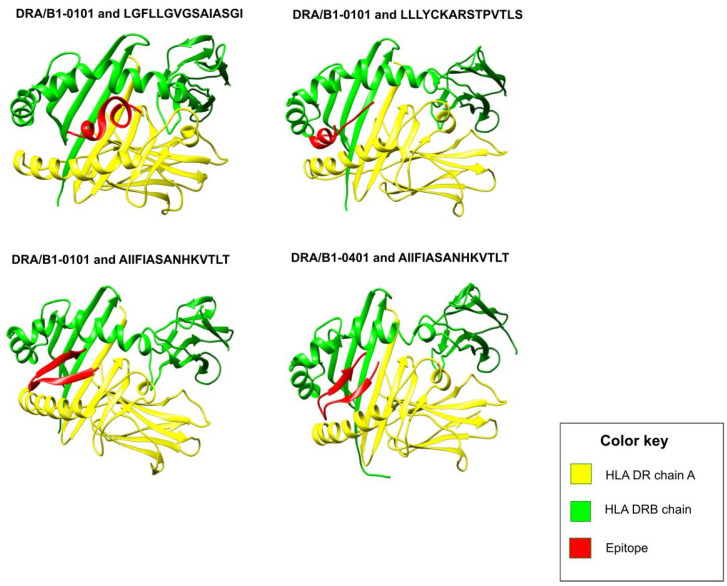
Figure showing structural interactions between the prioritized HLA II epitopes (colored in red) and cognate HLA II chains (colored in yellow and green). Out of all the HLA II epitopes, AIIFIASANHKVTLT (from the G glycoprotein) showed the best HADDOCK scores of −117.4 and −117.1 with HLA DRA/B1*0101 and HLA DRA/B1*0401, respectively. It can be observed that all the epitopes fit comfortably inside the binding groove of the HLA molecule (formed between the two HLA chains shown in yellow and green color, respectively), and thus are projected to interact strongly with their cognate HLA II allele.

**Figure 3 vaccines-10-01381-f003:**
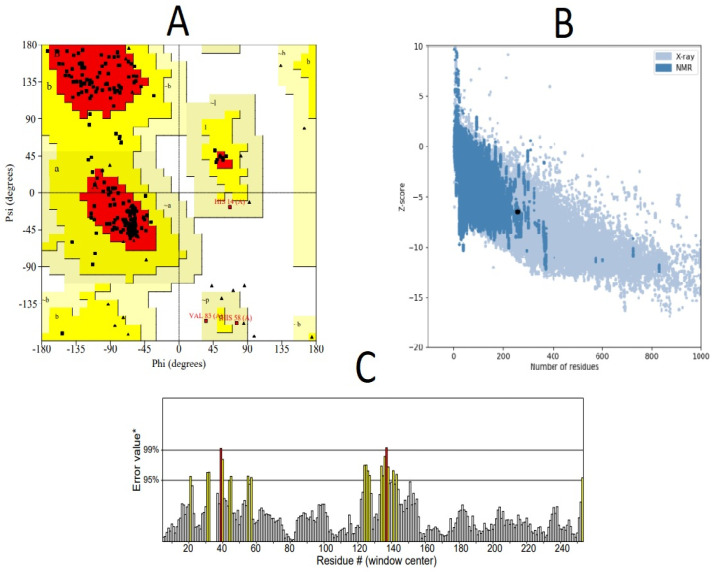
Structural evaluations of the modeled multi-epitope vaccine confirmed its high quality and stability using (**A**) Ramachandran plot analysis. It was confirmed that a total of 89% of the modeled residues resided in the most favored red region (indicated by capital letters A, B) whereas 9.5% and 1.5% residues resided in the additionally allowed bright yellow regions (indicated by small letters a, b) and generously allowed light yellow regions (indicated by ~a, ~b) respectively. Outlier regions are those having white background color. Glycine residues are indicated by triangles. (**B**) Prosa quality check confirmed that the modeled structure’s quality lies within the range of similar experimentally resolved NMR structures, with a Z-score of −6.47. (**C**) Stereochemical check by the ERRAT server revealed a high ERRAT score of 91.4286 that signifies that a good quality refined structure was obtained. On the error axis, two lines are drawn to indicate the confidence with which we can reject the regions that exceed that exceed that error value. Regions that have greater than 95% error are indicated by yellow color, whereas in rare cases residues having greater than 99% error values were found, indicated by red color.

**Figure 4 vaccines-10-01381-f004:**
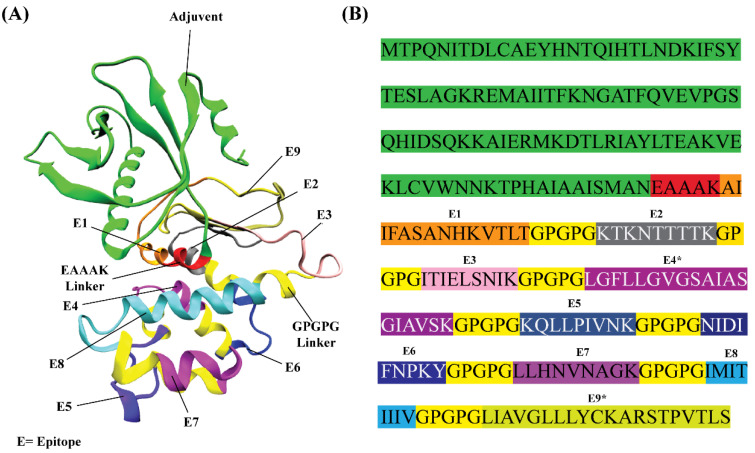
Constructed vaccine for RSV in the present study. (**A**) Modeled three-dimensional structure of vaccine construct. (**B**) Complete amino acid sequence of the vaccine construct. An adjuvant, cholera toxin subunit B, was placed towards the N-terminal of the construct. Specialized linkers (EAAAK and GPGPG) were added to separate the prioritized epitopes. Epitopes 4 and 9 are the merged CD4+ T-cell and CD8+ T-cell epitopes. These two epitopes are indicated by E4* and E9* (stars are added for easy identification).

**Figure 5 vaccines-10-01381-f005:**
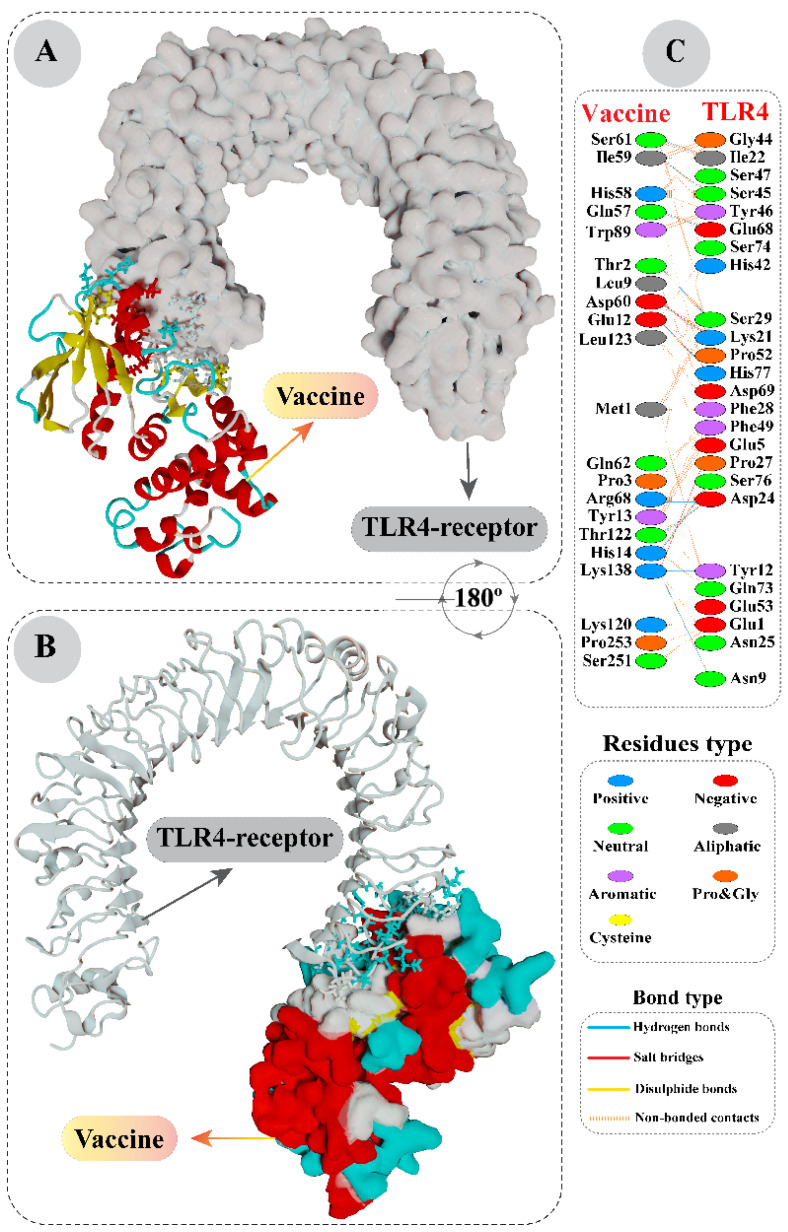
Protein-protein interaction of vaccine and TLR4. (**A**) Refined complex of multi-epitope with TLR4. Here, vaccine structure is shown in ribbon form, whereas the TLR4 structure surface has been represented. (**B**) Same complex as before, but after a 180-degree shift. However, TLR4 structure is shown in ribbon form, whereas the vaccine structure’s surface has been represented. (**C**) Structural analysis revealed numerous non-bonded contacts between vaccine and TLR4, as well as some notable hydrogen bonds and salt bridges. The type of amino acid residues involved in interactions have also been noted.

**Figure 6 vaccines-10-01381-f006:**
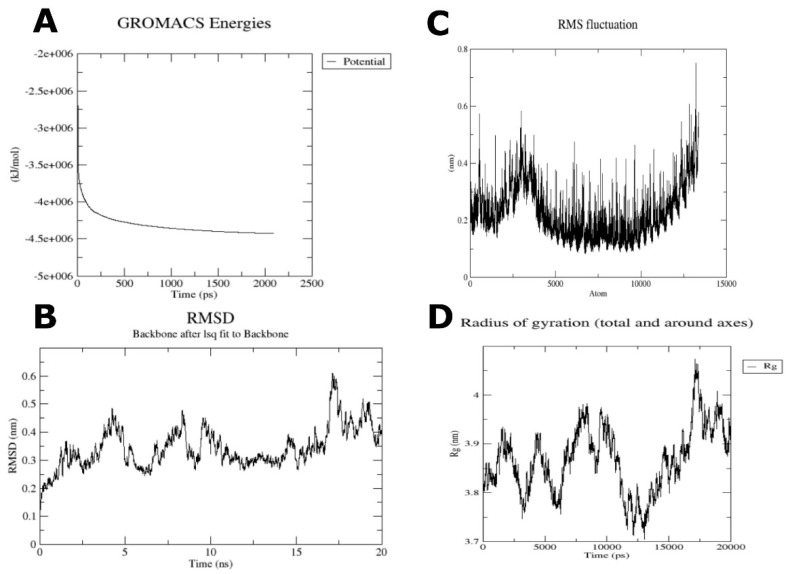
Some graphs obtained after molecular dynamics simulation on the vaccine-TLR4 complex. (**A**) Potential energy during simulation. (**B**) Comparison of backbone RMSD during MD production simulation. (**C**) RMSF of side chains during MD production simulation. (**D**) Radius of gyration with minimum fluctuations, showing compactness of structure during MD production simulation.

**Figure 7 vaccines-10-01381-f007:**
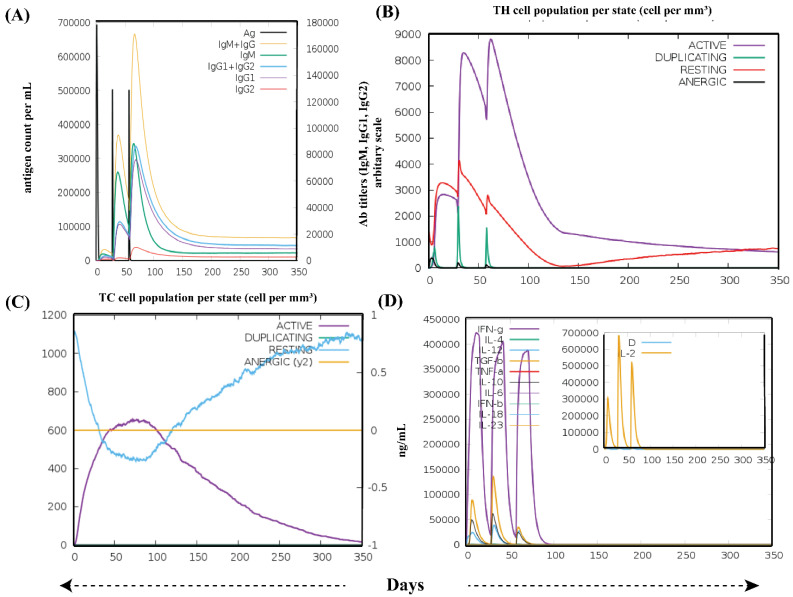
The immunostimulatory potential of the constructed vaccine after three injections of it. (**A**) Immunoglobulin production; (**B**) population/state of helper T-cell; (**C**) population/state of cytotoxic T-cells; (**D**) cytokines and interleukins production (smaller graph) with Simpson index. All units are stated in terms of cells per mm^3^ in three consecutive immunological responses.

**Figure 8 vaccines-10-01381-f008:**
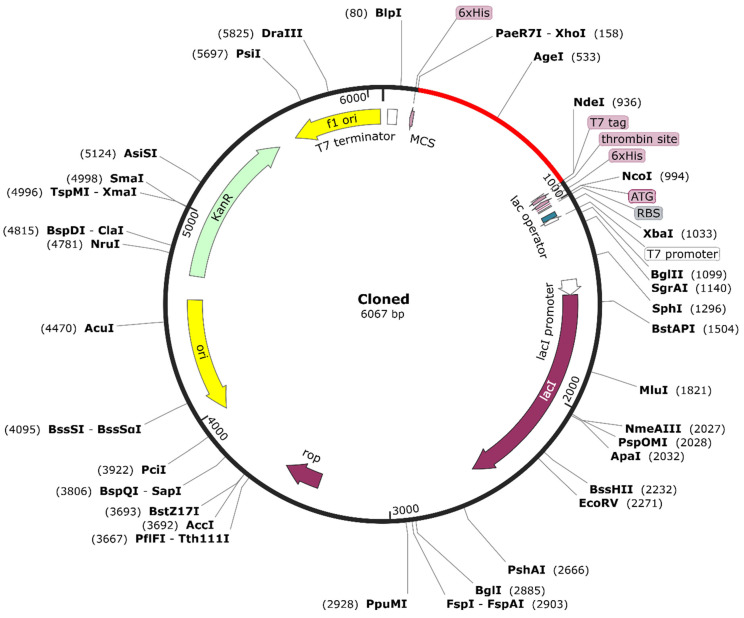
Simulated restriction cloning of the constructed multi-epitope vaccine. The multi-epitope vaccine construct’s codon-optimized sequence (shown in red color) was cloned in the pET-28a (+) expression vector (shown in black color) between the *Xho*I (158) and *Nde*I (936) restriction enzyme loci using SnapGene software. For efficient vaccine production, the designed construct can be expressed in *E. coli* (strain K12).

**Table 1 vaccines-10-01381-t001:** The overall attributes associated with the selected eight HLA I interacting T-cell epitopes. In epitope sequence, exposed residues are bold and italicized, while the rest are buried residues as predicted by the WESA tool.

Position	Epitopes	HLA Alleles	Protein	VaxiJen Score	Imunogenicity Score	IC50 Value
383–391	*N*I*D*IF*NPKY*	HLA-A*0101	Fusion glycoprotein	0.7826	0.11367	49.43
541–549	L*IAVG*LLL*Y*			0.8288	0.06096	30.13
525–533	IM*ITT*III*V*	HLA-A*0201	Fusion glycoprotein	0.5548	0.43542	77.62
512–520	LL*HNNAGK*	HLA-A*0301	Fusion glycoprotein	0.5354	0.09092	60.53
132–140	*KTKNTTTTK*		Attachment glycoprotein	0.8053	0.05327	54.83
57–64	*ITI*EL*NIK*	HLA-A*1101	Fusion glycoprotein	1.4507	0.04972	13.03
201–210	*KQ*L*PIVNK*			0.8386	0.13078	27.04
148–156	I*ASGA*V*SK*			0.8027	0.08501	36.48

**Table 2 vaccines-10-01381-t002:** Table showing features of the selected HLA II epitopes. In epitope sequence, exposed residues are bold and italicized, while other residues are buried (WESA tool).

Position	Epitope	HLA Alleles	Protein	VaxJen Score	IC50 Value
138–152	L*G*FLL*G*V*GSAIASGI*	DRB1*0101	Fusion glycoprotein	0.6271	37.33
546–560	LLLYC*KARSTPVT*L*S*	DRB1*0101		1.2472	2.75
	*A*IIFI*ASANHK*V*T*L*T*	DRB1*0101	Attachment glycoprotein	0.7845	40.18
58–72	*A*IIFI*ASANHK*V*T*L*T*	DRB1*0401			264.24

**Table 3 vaccines-10-01381-t003:** Table showing structural features of models generated by the Robetta server.

Model No.	Ramachandran Plot Analysis				ERRAT Score
	Most Favored Region	Additionally Allowed Regions	Generously Allowed Regions	Outlier Residues	
Model 1	89%	9%	1%	1%	89.9194
Model 2	84.50%	13.50%	1%	1%	96.6942
Model 3	87.50%	11%	0%	1.50%	91.5323
Model 4	89.50%	8%	1%	1.50%	90.1639
Model 5	89%	9.50%	1.50%	0%	91.4286

## Data Availability

All data generated or analyzed during this study are included in this published article.

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
