# Peer review of "Immunoinformatics-Aided Analysis of RSV Fusion and Attachment Glycoproteins to Design a Potent Multi-Epitope Vaccine"

_vaccines, 2022, doi:10.3390/vaccines10091381_

Round 1

Reviewer 1 Report

The manuscript entitled " Immunoinformatics-aided analysis of RSV fusion and glycoprotein regions to design a potent multi-epitope vaccine " by Hamza Arshad Dar, Fahad Nasser Almajhdi, Shahkaar Aziz and Yasir Waheed. In the MS, a multi-epitope subunit vaccine containing highly antigenic T-cell epitopes of the RSV fusion and glycoprotein regions was developed based on bioinformatics approaches. Below are some of my suggestions that could help readers and researchers in this field.

 --The author adds location or surface structural information of T cell epitopes in RSV structural proteins in MS, it will make the results more in-depth and conclusive.

--Glycosylation is very important for the antigenicity of some viral proteins. In this MS, in silico cloning ensured the increased expression of vaccine in Escherichia coli. Can this ensure the glycosylation or correct glycosylation of viral proteins in prokaryotic expression system (Escherichia coli)?

- Author need describe structural characteristics and function of the fusion and glycoprotein regions in MS. it would be worth briefly describing the important information to better understand design of multi-epitope vaccine.

-- It should be necessary to verify the effect of the vaccine through appropriate tests to accurately determine the accuracy of the method.

- Please review English writing and grammar throughout the manuscript.

Reviewer 2 Report

vaccines-1824794

The authors used various computational approaches to design a fusion and glycoprotein-based multi-epitope vaccine construct against respiratory tract infections caused by Respiratory Syncytial Virus (RSV).

1) Combine Figures 1 and 2 into one Figure

2) The color coding for the epitopes in Figures 1 and 2 is hard to read. Use some prominent colors; red?

3) Do peptides alone possess any defined fold? This information is needed to include and discussed in the draft to know whether the epitope binding to HLAs is causing structural transition or not.

4) Line 266; kDa, not kD

5) line 266: …lower than 110 kD molecular weight are more likely to show antigenic capabilities (add multi-epitope REF https://www.nature.com/articles/srep20613 )

6) Discuss the references related to epitope-based vaccines in general

https://academic.oup.com/bib/article/22/2/1309/6025506 (https://doi.org/10.1371/journal.pone.0119899

7) line 272: if the vaccine is hydrophobic, there is a high chance of protein aggregation. Then how is the construct going to be helpful?

8) Page8; the protein sequence should be displayed as a single, continuous sequence.

9) Figure 4: each figure needs to be discussed in the main text.

10) Figures 1-4: No details are given in the Figure captions.

11) What is the input to generate the structure models? More details are needed in the methods.

12) Authors should provide the PDB id and REF for TLR4.

13) Figure 5: A and C; this structural complex looks very transient. It is not providing any helpful information. Authors must present these well and discuss them. Maybe one structure is in surface presentation, the other in ribbon presentation, and vice versa.

14) The text font in figures should be increased.

15) Figure 7: Move the axis labels to the correct places. The text within the Figures overlapped with the images. Fix it.

 16) Authors overstated that the vaccine structure could be effective in humans.

Round 2

Reviewer 2 Report

Minor comments

1) "Therefore, the use of micelles is recommended to improve the interactions of the vaccine inside the polar environment of human body". 

Not sure how suitable the micelles are as they contain detergent moiety. Maybe say "...the use of a stable, lipid-based reconstitution system would be helpful to overcome protein's self-assembly.

2) Use courier new font and remove unnecessary space in protein and gene sequences in the SI file.

3) The Figures in the main text are at low resolution. They should be improved.

Author Response

1) "Therefore, the use of micelles is recommended to improve the interactions of the vaccine inside the polar environment of human body". 

Not sure how suitable the micelles are as they contain detergent moiety. Maybe say "...the use of a stable, lipid-based reconstitution system would be helpful to overcome protein's self-assembly.

Response: Thank you for your suggestion. We modified the sentence in the revised manuscript as per the reviewer’s suggestions.

2) Use courier new font and remove unnecessary space in protein and gene sequences in the SI file.

Response: We tried to include a continuous sequence inside word document but it always included hyphen at the end of a line. So, we converted both the gene and protein sequences into FASTA format and included FASTA formatted sequences in the Supplementary file. Also, we noticed that codon optimized sequence was not of 771 base pair length inside the supplementary file. So, we included the FASTA formatted 771 base pair long codon optimized sequence in the Supplementary File S2. Also, we updated the supplementary figures’ caption information inside the main manuscript file as well.

3) The Figures in the main text are at low resolution. They should be improved.

Response: High resolution figures were generated for use within the revised manuscript, either by using the source file directly or by adjusting the figure’s quality to make them visually appealing. We hope the updated figures will convey the required information clearly. Some figures and their captions were not placed in their correct positions so they were adjusted to correct their respective positions within the revised manuscript. We performed all these changes clearly by using ‘Track changes’ option. In case of any queries, please let us know.